# Dynamic mechanostereochemical switching of a co-conformationally flexible [2]catenane controlled by specific ionic guests

Yueliang Yao[1], Yuen Cheong Tse[1], Samuel Kin-Man Lai [1], Yixiang Shi[1], Kam-Hung Low [1] & Ho Yu Au-Yeung [1,2] ✉

Responsive synthetic receptors for adaptive recognition of different ionic guests in a competitive environment are valuable molecular tools for not only ion sensing and transport, but also the development of ion-responsive smart materials and related technologies. By virtue of the mechanical chelation and ability to undergo large-amplitude co-conformational changes, described herein is the discovery of a chameleon-like [2]catenane that selectively binds copper(I) or sulfate ions and its associated co-conformational mechanostereochemical switching. This work highlights not only the advantages and versatility of catenane as a molecular skeleton in receptor design, but also its potential in constructing complex responsive systems with multiple inputs and outputs.

While completely translating host–guest complementarity to the design and synthesis of a perfectly preorganized and rigid host is highly challenging, engineering flexibility into host framework enables conformational readjustment upon guest binding and strengthens the host–guest interactions[1]. However, structural flexibility often inevitably imposes an additional energy cost for host reorganization upon formation of host–guest complex[2]. A delicate balance of structural rigidity and flexibility is hence of critical importance in the design of molecular receptors, especially for multitopic hosts that need to change their conformations when binding to guests of vastly different steric and electronic features, and meanwhile minimizing the energy penalty for reorganization[3,4]. Selecting suitable covalent linkers of just enough flexibility and connecting binding motifs using these linkers at a right position are therefore key to a successful host design (Fig. 1). In this context, mechanical interlocking of rigid molecular components as mechanically interlocked molecules (MIMs) will be an alternative strategy to introduce structural flexibility into the host design[5,6]. In contrast to hosts with covalently linked binding motifs, the extent of preorganization in MIM-based hosts will be regulated by their co-conformational flexibility, which in turn can be fine-tuned by additional parameters that govern the nature of mechanical bond (e.g., ring tightness, strength of inter-component interactions)[7,8].

Moreover, the large-amplitude co-conformational changes unique to MIMs could allow the binding motifs to better sample the three-dimensional space around the guest while undergoing rearrangement, which serve to optimize the host–guest complementarity and allow the hosts to respond to changes in the external environment[9–12].

In this work, a heteroditopic [2]catenane composed of two interlocked macrocycles containing both bipyridine (bpy) and urea motifs for respective cation and anion binding is described. In addition to copper(I) ion that templates its synthesis, the host also displays an unexpectedly strong and selective binding to sulfate anion (log $K_1$ ~ 4.3 and log $K_2$ ~ 3.6 in 5% aqueous DMSO), despite no specific sulfate binding pocket has been included in the design of the catenane host. Although the covalent backbone of the individual macrocycles is relatively rigid, the [2]catenane is co-conformationally flexible that the interlocked rings can undergo a "180°-turn" to switch from the achiral, copper(I)-bound co-conformer to the mechanical chelating sulfate complex in a chiral co-conformation. In addition to the significance in the transport[13,14], sensing[15–18], delivery, and recovery of ionic species and resources[19–21], this work also highlights mechanical interlocking as an efficient element in designing molecular receptors with versatile structures, properties and guest binding behavior.

[1]Department of Chemistry, The University of Hong Kong, Hong Kong, China. [2]State Key Laboratory of Synthetic Chemistry and HKU-CAS Joint Laboratory on New Materials, The University of Hong Kong, Hong Kong, China. ✉e-mail: hoyuay@hku.hk

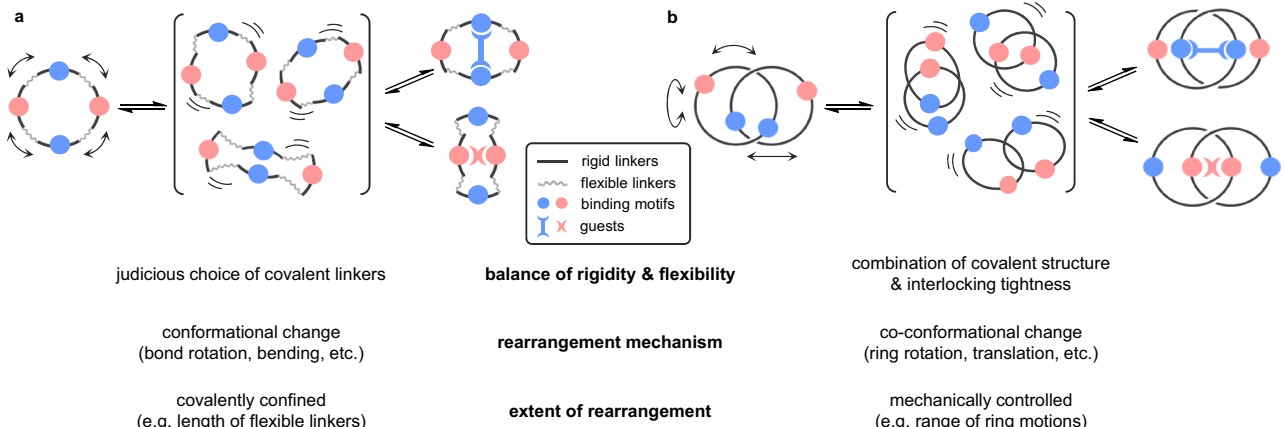

**Fig. 1 | Designs and features of partially flexible hosts. a** Schematic illustration of a macrocyclic host with rigid and flexible covalent linkers. **b** Schematic illustration of a [2]catenane host with mechanically flexible but covalently rigid macrocycles.

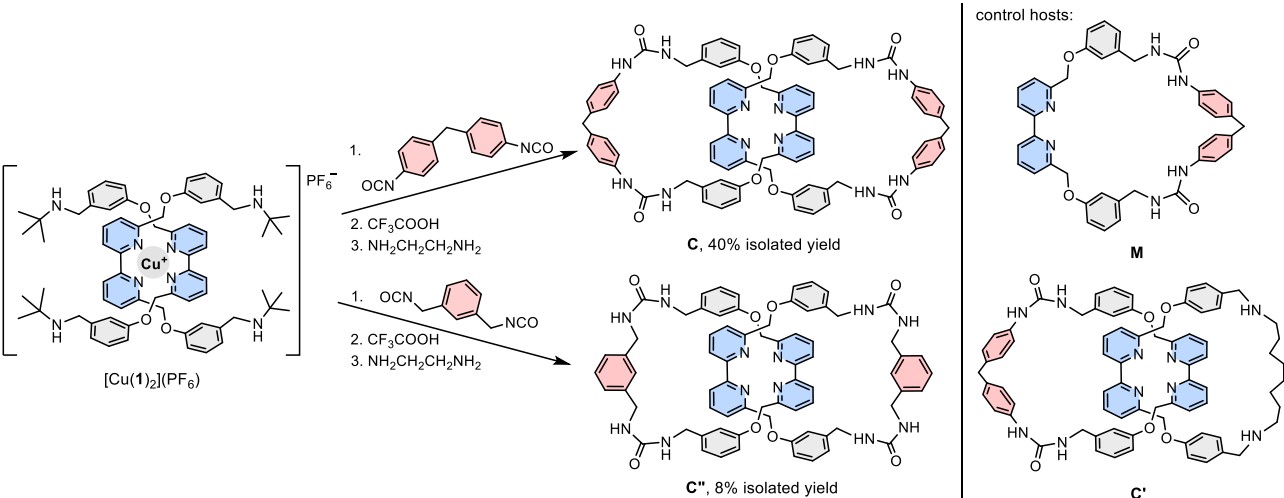

**Fig. 2 | Synthesis of tetra-urea catenanes and structures of control bis-urea hosts.** The tetra-urea [2]catenanes **C** and **C″**, and bis-urea control macrocycle **M** and [2] catenane **C′** were synthesized from dynamic urea formation from the corresponding *t*-butylamine and isocyanate precursors.

## Results and discussion

To obtain catenane-derived receptors, the Cu⁺-templated, dynamic urea formation from building blocks **1** and **2** containing respectively *t*-butylamine and isocyanate functional groups was explored to simultaneously introduce the cation binding bipyridine (bpy) and anion binding urea groups during the catenane synthesis[22,23]. Catenane **C** is obtained in three simple steps without any chromatographic purification (Fig. 2). By virtue of the strong Cu⁺-bpy template and the effective ring-closing from the error-checking dynamic urea formation, the catenane topology was efficiently created. Simple acid treatment removed the *t*-butyl group after the catenane formation, which locked the dynamic urea and activated their anion binding potential[24]. Catenane **C** was then obtained by using ethylenediamine to extract the Cu⁺ template followed by aqueous washing[25]. Control [2]catenane **C′** and macrocycle **M** featuring only one bis-urea bpy-based macrocycle were also prepared to evaluate the effects of mechanical interlocking on the binding (Fig. 2 and Supplementary Figs. 1 and 2). Of note, because of the different possible ways of connecting the reactive ends of [Cu(**1**)₂]⁺ when diisocyanates of different structural features are used, formation of the catenane topology is not necessarily efficient, and attempts to synthesize related catenanes were either unsuccessful or only gave the target [2]catenane in a low yield. For example, the related catenane **C″** was obtained only in 8% yield along with other non-

interlocked products by following the same synthetic procedure (see Supplementary Fig. 3 for more details)[26].

HR-ESI-MS spectrum of **C** showed molecular ion peaks at *m/z* 1375.5458 [**C** + Na]⁺ and 677.2873 [**C** + 2H]²⁺ that are consistent to the expected molecular formula $C_{82}H_{72}N_{12}O_8$ of the catenane, and the interlocked nature is evidenced by the direct fragmentation of the singly charged molecular ion peak to the constituting macrocycle at *m/z* 677.2837 [**M** + H]⁺ in the MS/MS experiment (Supplementary Fig. 5). The ¹H NMR spectrum (600 MHz, 298 K, DMSO-*d₆*) of **C** showed that the catenane adopts a highly symmetrical structure with three $H_{bpy}$ and two $H_{dpm}$ resonances, suggesting that the two interlocked rings are equivalent and the [2]catenane possesses two $C_2$ axis. The two urea NH were found at 6.52 ppm and 8.25 ppm, which are similar to other reported urea-based macrocycles[23,27]. Only slight spectral changes were observed when the temperature was increased from 298 K to 358 K (Supplementary Fig. 26), which may be explained by a rapid co-conformational exchange of the interlocked host. Attempt to obtain the low-temperature NMR spectrum of **C** in other solvents such as CDCl₃ failed due to its poor solubility. Comparing the ¹H NMR spectra of **C** and the non-interlocked macrocycle **M**, proton resonances are generally more upfield shifted in the catenane due to a stronger shielding effect as a result of mechanical interlocking (e.g. $\Delta\delta$ - 0.52−0.61 ppm for $H_{bpy}$ and 0.30−0.42 ppm for $H_{dpm}$).

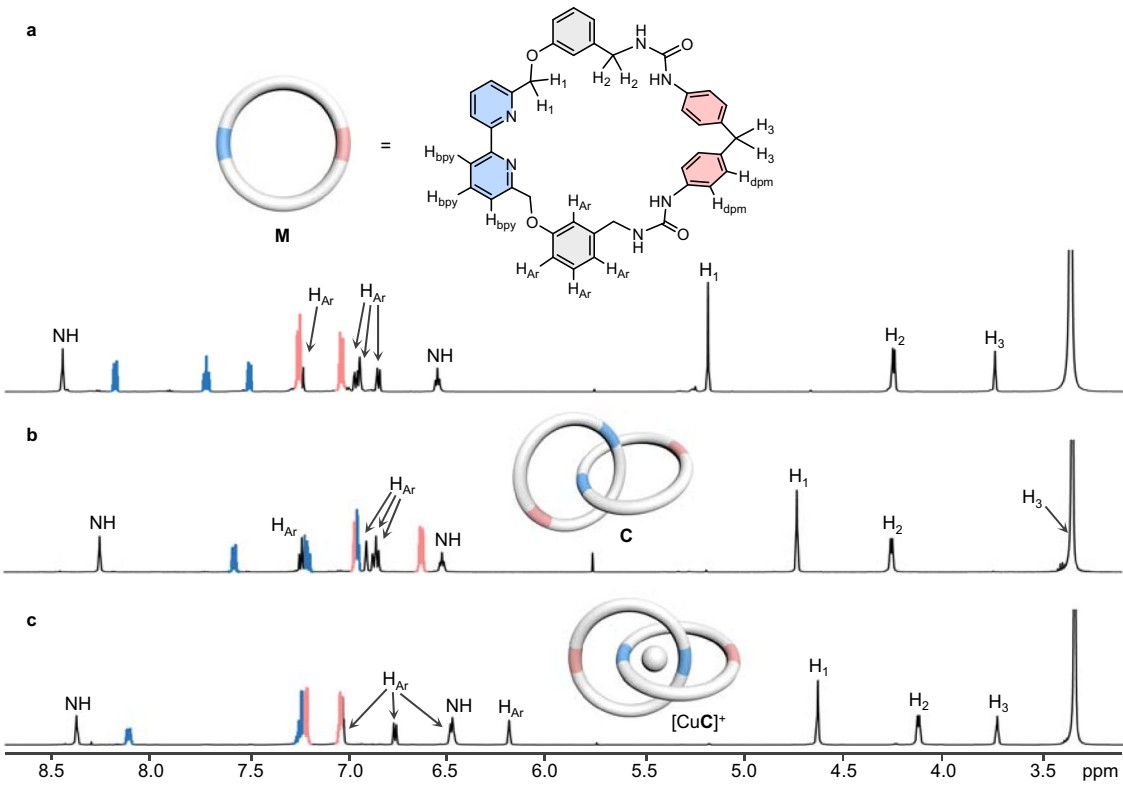

**Fig. 3 | $^1$H NMR studies of the [2]catenane.** Partial $^1$H NMR spectra (600 MHz, DMSO-$d_6$, 298 K) of **a** macrocycle **M**, **b** catenane **C**, and **c** Cu(I) complex [Cu**C**](PF$_6$). Aromatic signals from bpy and dpm units are highlighted in blue and pink respectively.

Similar to most other metal-templated catenanes, the bis(bpy) cavity crafted in **C** during the templated synthesis could be complementary for cation binding[11,28–30]. As expected, addition of [Cu(MeCN)$_4$](PF$_6$) to a solution of **C** led to the formation of a species with a new UV-Vis absorption at 450 nm characteristic to the Cu$^+$-to-bpy MLCT and a $^1$H NMR spectrum consistent to that of [Cu**C**]$^+$ (Supplementary Fig. 30 and 36). The 1:1 Cu(I) complexation was found to be in slow exchange on the NMR timescale, and no further spectral changes were observed after one equivalent of Cu$^+$ ion had been administered. In the $^1$H NMR spectrum of [Cu**C**](PF$_6$), resonances of the *meta*-substituted aryl spacer signals H$_{Ar}$ are significantly upfield shifted when compared to that of the free host (Fig. 3), and NOE cross peaks between H$_{Ar}$ and H$_{bpy}$ were also found in the 2D NOESY spectrum (Supplementary Fig. 28). These spectral features are consistent with the reinforced π-stacking interactions between the aromatic units as a result of the Cu$^+$-bpy coordination[22]. Slight chemical shift changes were found for the urea NH, H$_1$, and H$_2$, and a more significant downfield shift by *ca.* 0.4 ppm was noticed for the diphenylmethylene (dpm) H$_3$. Because of the interlocked structure, the dpm linkers are likely perpendicularly oriented with respect to the bpy of the other macrocycles, hence placing H$_3$ in the deshielding region of the bpy in the Cu(I) complex.

While the Cu$^+$ binding may not be surprising, the bpy-derived [2] catenane showed a very different cation selectivity than those closely related bis(phenanthroline) catenanes. Although catenanes with a bis(phenanthroline) core have been shown to form complexes with a range of other cations[31,32], no obvious changes in the $^1$H NMR and UV-Vis spectra were observed when Li$^+$, Na$^+$, Fe$^{2+}$, Co$^{2+}$, Ni$^{2+}$, Zn$^{2+}$, and Cd$^{2+}$ ions were added to a DMSO solution of **C** (Supplementary Fig. 30 and 31). The lack of apparent binding of other metal ions to **C** is likely due to a slow coordination kinetics due to the (co)conformational flexibility of the bpy and the catenane, effects from the counter ions and solvent, as well as a mismatch in the coordination geometry. In DMSO, no coordination of 6,6'-dimethyl-2,2'-bipyridine to Zn$^{2+}$ (as ClO$_4^-$ salt)

was observed (Supplementary Fig. 38), and similar kinetic effects have indeed been reported for the direct metalation of a bpy-derived molecular knot[33]. The addition of 1 eq. of Cu$^{2+}$ led to proton signals that are assignable to [Cu**C**]$^+$ and a new absorption peak at 450 nm in the $^1$H NMR and UV-Vis spectra respectively, showing that the Cu$^{2+}$ has been reduced with the subsequent formation of the Cu(I) catenane[34]. On the other hand, broadening and gradual shifting of the proton resonances of **C** were observed when up to 2 eq. of Ag$^+$ ion was added, suggesting the formation of a new, fast exchanging Ag$^+$-associated species in DMSO. Yet, the Ag$^+$ binding is not as strong as that of Cu$^+$, and the Ag$^+$-associated species can be converted to [Cu**C**]$^+$ when a stoichiometric amount of Cu$^+$ was introduced (Supplementary Fig. 37). The fast exchange of the Ag(I) species may be unusual if the Ag$^+$ ion is fully encapsulated in the catenane cavity[35,36], and more extensive structural analysis and studies will be warranted to fully understand the impact of the catenand effect on the coordination properties of these bpy-derived MIMs on different metal ions.

The incorporation of four hydrogen bonding urea motifs in the macrocycle backbone is envisaged to endow the [2]catenane **C** with potent anion binding capability. To investigate its anion recognition behavior, $^1$H NMR anion titration experiments were conducted, in which aliquots of different anions (Cl$^-$, Br$^-$, I$^-$, NO$_3^-$, HSO$_4^-$, SO$_4^{2-}$, CH$_3$COO$^-$, H$_2$PO$_4^-$) as their tetrabutylammonium (TBA) salts were added to the catenane solutions in DMSO-$d_6$ (Supplementary Fig. 40). Only negligible spectral changes were observed when Br$^-$, I$^-$, NO$_3^-$, HSO$_4^-$ were added, suggestive of no significant binding to these anions. In stark contrast, the addition of sulfate resulted in a drastic downfield perturbation to the urea NH signals ($\Delta\delta = 1.56$ ppm and 2.22 ppm over 2 eq. of sulfate), suggesting the oxoanion is bound via the formation of urea NH hydrogen bonds (Fig. 4). Further addition of sulfate (>2 eq.) did not induce further chemical shift changes, suggesting that the urea anion binding sites have been saturated. Downfield shifts of urea proton signals, in a less pronounced magnitude were also observed upon the addition of 10 eq. of Cl$^-$, H$_2$PO$_4^-$ and

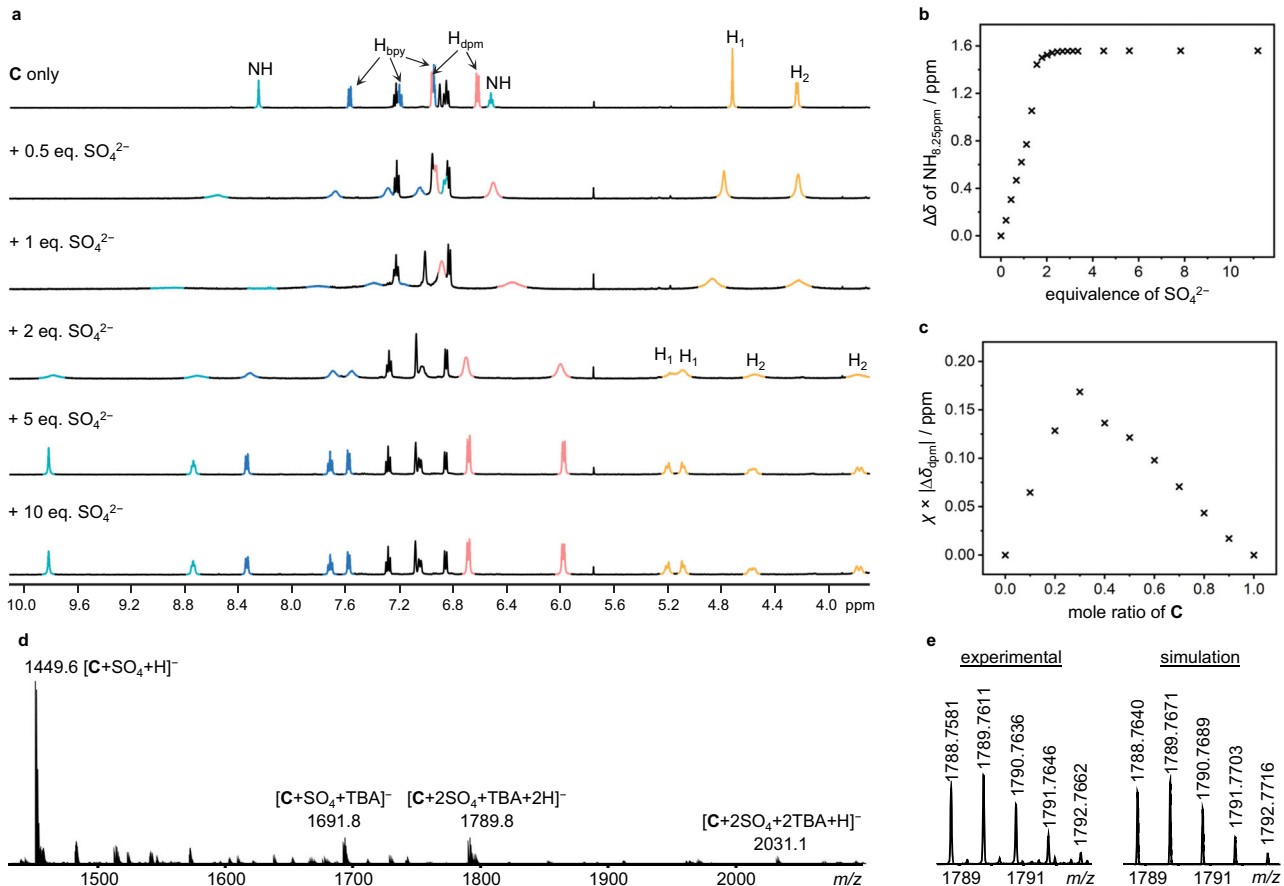

**Fig. 4 | Characterization of sulfate binding to catenane C. a** Partial $^1$H NMR (500 MHz, DMSO-$d_6$, 298 K) spectra of **C** in the presence of increasing amount of TBA sulfate (signals from urea NH, bpy, dpm and methylene protons are highlighted in green, blue, pink and yellow respectively). Methylene protons H$_1$ and H$_2$ become diastereotopic in the presence of 2 eq. or more of sulfate. **b** Change in chemical shift of the urea NH proton originally at 8.25 ppm. **c** Job plot of the binding showing a 1:2 host–guest stoichiometry. **d** HR-ESI-MS (-ve) spectrum obtained from a sample of **C** in the presence of 10 eq. of TBA sulfate. **e** Isotopic distribution of the mass signal at *m/z* 1789.8.

CH$_3$COO$^-$, indicating that the catenane also binds to these anions but with a weaker affinity. The anion binding stoichiometry was revealed by Job plot (Supplementary Figs. 41 and 42), and the 1:1/1:2 host–guest anion association constants were determined from the titration data and are summarized in Table 1.

In particular, the catenane was found to be selective to sulfate with a 1:2 stoichiometry and binding constants of >10$^5$ M$^{-1}$ in DMSO-$d_6$ for both the first and second sulfate association. Encouraged by the strong and selective binding of sulfate anion in DMSO-$d_6$, $^1$H NMR titration experiment was carried out in the more competitive 5% D$_2$O in DMSO-$d_6$. For comparison, the sulfate association constants of the macrocycle **M**, and the control [2]catenane **C**′ consisting of only one bis-urea-based bpy macrocycle, were also determined. Both

macrocycle **M** and catenane **C**′ displayed comparable 1:1 host–guest binding to sulfate anion with *K* ~ 400 M$^{-1}$. In contrast, the tetra-urea catenane **C** demonstrated significantly stronger 1:2 host–guest association to the oxoanion, with $K_1 = 2.1 \times 10^4$ M$^{-1}$ and $K_2 = 3.6 \times 10^3$ M$^{-1}$ at 298 K. The negative cooperativity ($K_1/K_2 \approx 5.8$) of the first and second sulfate binding is likely due to the electrostatic repulsion between the two anions within the neutral catenane[37]. Van't Hoff analysis shows the binding is driven by enthalpy with a negative entropy change ($\Delta H = -23$ kcal mol$^{-1}$; $\Delta S = -41$ cal mol$^{-1}$ K$^{-1}$), which could be explained by the decrease in the (co)conformational flexibility of the catenane upon guest binding.

The sulfate complex can also be directly observed by HR-ESI-MS, in which peaks for the 1:1 and 1:2 sulfate complexes at *m/z* 1449.5154 [**C** + SO$_4$ + H]$^-$, 1691.7937 [**C** + SO$_4$ + TBA]$^-$ and 1789.7611 [**C** + 2(SO$_4$) + 2H + TBA]$^-$, with isotopic distributions corresponding to the respective molecular formula, were found in the MS spectrum (Fig. 4). These findings suggest the mode of sulfate binding by **C** features a mechanical chelation of a sulfate anion by two urea groups from the two interlocked rings, and more importantly highlights the critical role of the mechanical flexibility for efficient, large-amplitude co-conformational rearrangement of the binding motifs despite the rigidity of the covalent macrocyclic skeleton.

Apart from the strong binding in competitive aqueous environment, sulfate binding of **C** is also characterized by a distinctive structural change of the host. In addition to the downfield shifting of the urea NH, the $^1$H NMR spectrum (500 MHz, DMSO-$d_6$, 298 K) of **C** in the presence of 10 eq. SO$_4$$^{2-}$ also showed that the H$_{bpy}$ resonances are

**Table 1 | Anion binding constants of catenane C and control compounds C′ and Mª**

| Anion binding of C (DMSO-$d_6$) | | | | Sulfate binding (5% D$_2$O/DMSO-$d_6$) | | | |
|---|---|---|---|---|---|---|---|
| **Anion** | **SO$_4$$^{2-}$** | **AcO$^-$** | **H$_2$PO$_4$$^-$** | **Cl$^-$** | **Host** | **C** | **C′** | **M** |
| $K_1$ (M$^{-1}$) | >10$^5$ | 7700 | 3000 | 160 | $K_1$ (M$^{-1}$) | 21000$^b$ | 370 | 360 |
| $K_2$ (M$^{-1}$) | >10$^5$ | 370 | 390 | 3 | $K_2$ (M$^{-1}$) | 3600 | / | / |

ªAnion binding constants $K_1$ and $K_2$ of tetra-urea [2]catenane **C** and bis-urea control compounds (**C**′ and **M**) were calculated using Bindfit (http://app.supramolecular.org/bindfit/) with 1:1 or 1:2 host–guest binding models[70]. Errors (±) are all <10% unless otherwise noted. All anions were added as TBA salts. [Host]$_0$ = 1 or 2 mM. *T* = 298 K. $^b$Error = 18.4%.

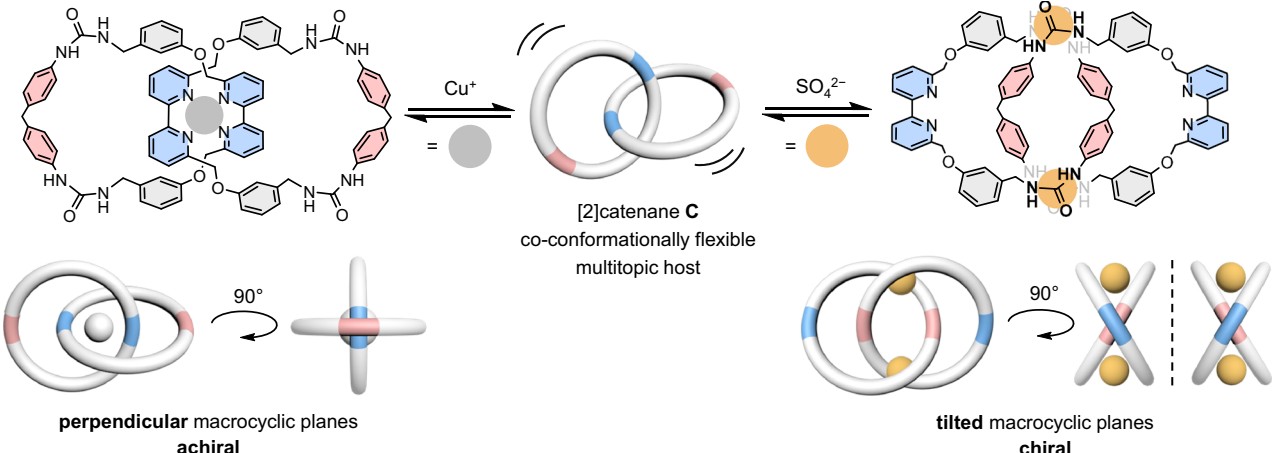

**Fig. 5 | Co-conformational switching of catenane C upon selective ion binding.** Co-conformational exchange of **C** involving a 180°-turn of the macrocycles upon guest binding to produce achiral Cu(I) and chiral sulfate complexes. Schematic illustrations of complexes in different perspectives are provided (blue: bpy; pink: dpm; gray ball: Cu⁺; yellow ball: $SO_4^{2-}$).

downfield shifted to 8.34 ppm, 7.72 ppm, and 7.58 ppm, while that of the $H_{dpm}$ are more upfield at 6.69 ppm and 5.98 ppm when compared to that of the free host **C** and macrocycle **M**, suggestive of a respectively weaker and stronger shielding effect for the bpy and dpm units in the sulfate complex. A significant upfield shift of the dpm $CH_2$ ($H_3$) was also observed in the sulfate complex, indicating that **C** adopts a co-conformation in which the bpy are more exposed at the catenane exterior, and concomitantly the dpm are buried in the catenane center upon sulfate binding[38]. Compared to the co-conformation in [Cu**C**]⁺ in which the bpy and dpm units locate respectively at the central and peripheral positions of the catenane, the two interlocked macrocycles have each undergone a "180°-turn", and the bpy and dpm exchanged their relative positions in the sulfate complex. Furthermore, the 2D NOESY spectrum of **C** obtained in the presence of 10 eq. $SO_4^{2-}$ showed NOE cross peaks between $H_{dpm}$ and $H_{Ar}$ protons (Supplementary Fig. 29), which is consistent to the interlocked macrocycles rotated inside-out, such that the dpm units are now in a closer proximity to the *meta*-substituted aryl spacer of the other macrocycle. Noteworthily, literature examples of sulfate-selective receptors frequently invoke a hydrophobic effect from host encapsulation of sulfate anions, which shields the highly hydrophilic anion and assists its desolvation in aqueous environment[27,39–46]. In contrast, sulfate anion induces a large-amplitude co-conformational change in the interlocked backbone of **C**, bringing the urea donors in the two macrocycles in proximity for convergent sulfate binding. The high flexibility and ability to undergo large-amplitude co-conformational change thus enable the catenane host to adapt to ionic guests of different size, charge, geometry and binding stoichiometry (i.e., 1:1 binding of monocationic, spherical Cu⁺ and 1:2 binding of dianionic, tetrahedral $SO_4^{2-}$).

The anion binding of **C** is also studied by DFT modeling, and the computed association constants were found comparable to our experimental data (Supplementary Table 1). Consistent to the NMR structural studies, the energy-minimized structure of the 1:2 sulfate complex features a mechanical chelate connected via NH···O hydrogen bonds between the urea and the anion, and the dpm units are buried in the center of the catenane (Supplementary Fig. 63). Close contacts between the anions and other parts of the catenane were also observed in these simulated anion complexes, suggesting that all the urea hydrogen bonds and these other interactions, albeit some of which are less obvious, are collectively responsible for the overall stability of the anion complexes. The anion selectivity is therefore a result of the delicate balance of all these non-covalent interactions involving the specific anion and the different catenane (co)conformations.

Sulfate binding also leads to a change in the stereochemistry of **C**. While the time-averaged structures of the free host and Cu(I) complex [Cu**C**]⁺ display no chiral feature in our spectroscopic studies, diastereotopic splitting of the methylene singlets $H_1$ and $H_2$ to a pair of spin-coupled doublets can be observed in the presence of 10 eq. $SO_4^{2-}$, showing that the sulfate ions stabilize a specific chiral co-conformer of **C**. Weakening the sulfate association by adding $D_2O$ to the NMR sample in DMSO-$d_6$ led to the coalescence of $H_1$ and $H_2$ doublets (Supplementary Fig. 58), and in the presence of 5% $D_2O$ or above, the resulting ¹H NMR spectrum essentially showed no chiral feature although significant sulfate binding can still be measured as discussed previously. Variable temperature ¹H NMR studies of the sulfate complex showed the diastereotopic doublets of $H_1$ coalesce at 318 K, corresponding to a racemization barrier of 15.7 kcal·mol⁻¹ in DMSO-$d_6$ (Supplementary Fig. 57). These findings not only confirm that sulfate selects and binds strongly to a chiral co-conformer of the catenane host, but also demonstrate a rare example in which the extent of chiral property expression can be controlled by the specific condition of host–guest binding. For both the Cu(I) and sulfate complexes of **C,** ¹H NMR analysis showed that the two interlocked macrocycles are equivalent, and both complexes are $C_2$ symmetric with respect to the catenane molecular axis with only three bpy and two dpm aromatic signals. Chirality of the sulfate complex is therefore explained by a tilted orientation of the two macrocyclic planes as a result of the rocking of the macrocycles being arrested by the sulfate binding, and such structure has been referred as being mechanically helically chiral (Fig. 5)[47–54]. On the other hand, the tetrahedral Cu⁺-bpy coordination brings the bpy to the catenane center and enforces a perpendicular orientation of the macrocyclic planes to give an achiral structure. Other co-conformations of **C** are unlikely to be $C_2$ symmetric although some of them are also chiral[55,56]. Finally, the chiral structure induced by sulfate binding is found to be characteristic to **C**, and the related urea catenane **C″** showed no chiral spectral feature although the interlocked host also displayed a strong sulfate binding in DMSO-$d_6$ (Supplementary Fig. 54).

Further structural details of the sulfate complex are revealed by single crystal X-ray diffraction analysis (Fig. 6). Single crystals of the sulfate complex were obtained by slow vapor diffusion of diisopropyl ether into a DMF solution of **C** in the presence of 10 eq. $TBA_2SO_4$. The sulfate complex crystallized in the monoclinic $P2_1/n$ space group with each asymmetric unit consists of one catenane, one sulfate, two TBA ions and four DMF molecules. Consistent with the NMR structural studies, the catenane was found to adopt a co-conformation in which the dpm and bpy locate respectively in the catenane center and

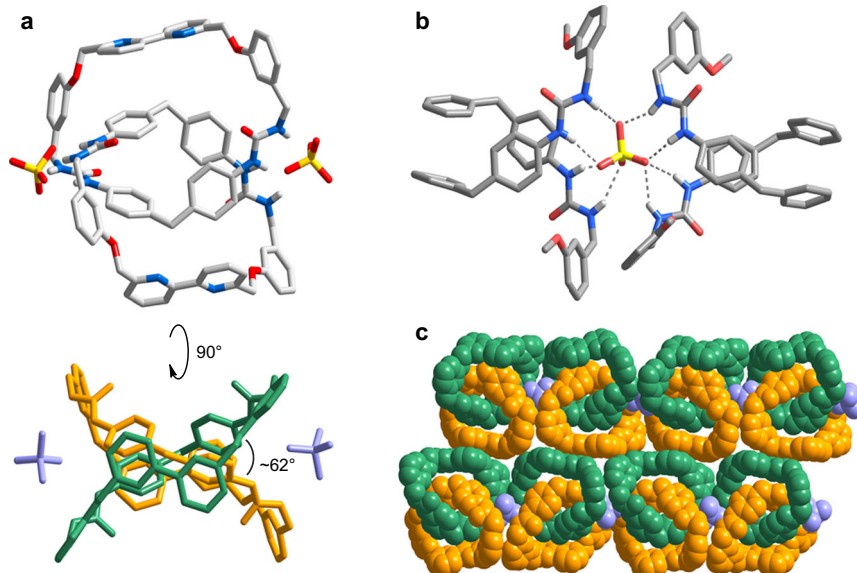

**Fig. 6 | Solid-state structures of the sulfate complex of C. a** X-ray crystal structure showing the sulfate catenane complex in different perspectives. Color code: gray, C; blue, N; red, O; white, H; yellow, S. **b** Hydrogen bonding interactions between the sulfate and urea groups from two catenanes. **c** Arrangement of the enantiomeric co-conformers. The two macrocycles of **C** are colored in green and orange respectively, and sulfate anions are colored in purple. TBA ions and DMF molecules are omitted for clarity.

periphery, and an average interplanar separation of 4.1 Å was found between the dpm units from the two macrocycles. The bpy units are also found to adopt a *trans* configuration that the two pyridine donors are pointing toward opposite directions, probably as a result of relieving any potential ring strain in the interlocked macrocycle. Eight N–H···O hydrogen bonds at a distance of 2.755–3.133 Å were identified between each of the sulfate and four urea groups from two different interlocked rings of two molecules of **C**, confirming the mechanical chelating mode of the sulfate binding. The chiral co-conformation is evidenced by the tilted arrangement of the macrocyclic planes at an angle of ~62°. The crystal sample is overall racemic and the enantiomeric co-conformers of opposite chirality are bridged by a sulfate ion in an alternate fashion to result in infinite 1D chains, which further align with adjacent strands at a distance of ~4.1 Å (Supplementary Fig. 65).

The guest-controlled, reversible co-conformational mechanostereochemical switching of **C** was further demonstrated by the sequential introduction of $Cu^+$ and $SO_4^{2-}$, as well as the competitive $CN^-$ and $Ba^{2+}$ ions. As shown in Fig. 7, addition of 4 eq. of $TBA_2SO_4$ to a 1 mM DMSO-$d_6$ solution of **C** resulted in the formation of the 1:2 sulfate complex as shown by the downfield shifted urea NH, as well as the diastereotopic $H_1$ and $H_2$ resonances. This chiral, sulfate-bound form of **C** can be made co-conformationally flexible again by addition of $Ba(OTf)_2$ that led to the formation of $BaSO_4$ and reversed the spectral changes, with a resulting $^1H$ NMR spectrum essentially the same as that of the initial free host **C**. Further switching to the achiral form of the $Cu^+$ complex was achieved by addition of $[Cu(MeCN)_4](PF_6)$, which produced a $^1H$ NMR spectrum with the $H_{bpy}$ and $H_{Ar}$ signals characteristic to a $Cu^+$-bis(bpy) in an overall achiral structure. The guest-free catenane was regenerated again by adding $[(Me_4N)(CN)]$ that competitively extracted the $Cu^+$ ion from the catenane, which could then switch on its sulfate binding ability and gave the chiral sulfate chelate after the introduction of a second batch of $TBA_2SO_4$. The successful use of four ionic guests of different charges, geometry, and properties to control a multitopic host to give two different stereochemical outcomes, by virtue of the exceptional co-conformational flexibility of the mechanical bond, hence highlights the unique potential of exploiting the topologically non-trivial catenane as a receptor skeleton in the design and development of diverse classes of smart and responsive molecular hosts. Of note, although ion-

controlled multi-state switching in MIM-based molecular switches has been described, the binding cavities in these examples are usually pre-designed with known binding and structural features favorable for the specific guests[57-62]. Our discovery that catenane **C** can selectively form a strong mechanical chelate with sulfate in an unexpected, chiral co-conformation thus demonstrates the distinctive advantage of combining rigid macrocycles and a co-conformationally flexible, mechanically interlocked molecular framework as next-generation receptors with the ability to undergo facile host–guest optimization while maintaining a high degree of preorganization.

In summary, a novel and efficient approach to prepare heteroditopic [2]catenane has been developed, exploiting the metal cation template-directed strategy and dynamic urea formation as ring-closing reaction. Extensive binding studies revealed that in addition to encapsulating the spherical $Cu^+$ cation at the core via mechanical chelation of the two bpy ligands in an overall 1:1 stoichiometry, the [2]catenane is also capable of strong and selective binding to the tetrahedral $SO_4^{2-}$ anion at the periphery mediated by urea hydrogen bonding formation in a 1:2 host–guest fashion. $^1H$ NMR analysis and X-ray crystallographic study showed the interlocked macrocycles are related by a "180°-circumrotation" in the Cu(I) and sulfate complexes, and saliently, whilst the former remains achiral, the catenane·sulfate complex is chiral with a tilted orientation of the macrocyclic planes. These findings not only highlight the advantages of using co-conformationally flexible catenane hosts for facile adaption and rearrangement of binding motifs to accommodate guests of various structural and electronic features, but also demonstrate a unique example in which the realms of dynamic mechanostereochemistry and host–guest chemistry characteristic to MIMs intersect, where new and unexplored opportunities may arise.

## Methods
Full experimental details and characterization of compounds can be found in the Supplementary Information.

### Materials
All reagents (purity ≥98% unless otherwise noted) were purchased from commercial suppliers (J&K, Sigma-Aldrich, TCI, Merck, Aladdin, Energy, Macklin, and Dkmchem) and used without further purification

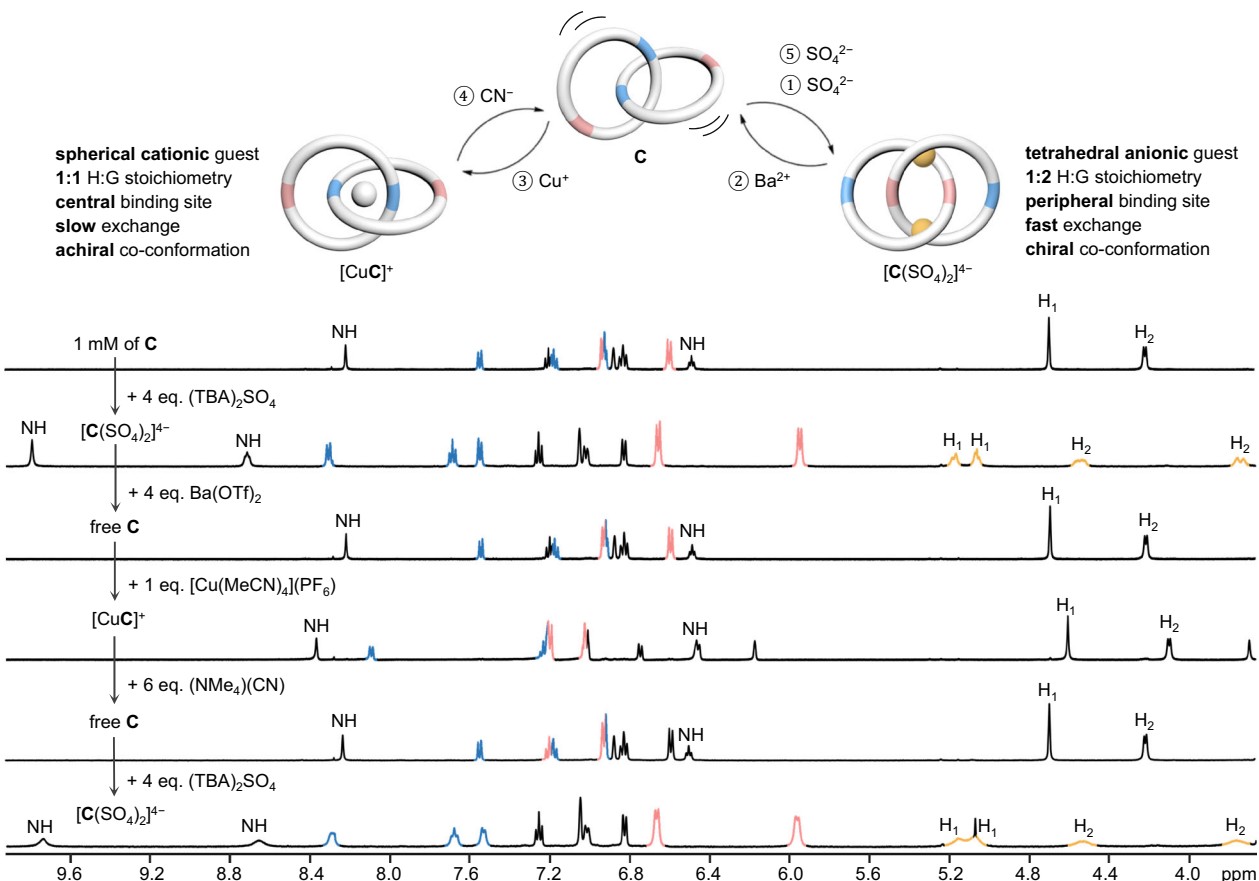

**Fig. 7 | Reversible achiral-chiral switching of C.** Upper: Schematic illustration of co-conformations of **C** switching between achiral and chiral by sequential addition and removal of sulfate and Cu(I) guests. Bottom: Partial $^1$H NMR (500 MHz, DMSO-$d_6$, 298 K) spectra of **C** and species obtained after additions of different salts. Signals from $H_{bpy}$ and $H_{dpm}$ are highlighted in blue and pink respectively. Diastereotopic splitting of $H_1$ and $H_2$ in the sulfate complex is highlighted in orange.

unless otherwise noted. All the solvents were of analytical grade (ACI Labscan and DUKSAN Pure Chemicals).

## General procedures for [2]catenane synthesis

Under an argon atmosphere, a mixture of bis-*t*-butylamine **1** (1 eq.) and [Cu(MeCN)$_4$](PF$_6$) (0.5 eq.) or bis-*t*-butylamine **1** (1 eq.), macrocycle **3** (1.1 eq.) and [Cu(MeCN)$_4$](PF$_6$) (1.1 eq.) in MeCN/CHCl$_3$ (*v/v* 1:1) was heated at 50 °C for 1 h to 3 h. The red reaction mixture was cooled to room temperature and filtered to remove insoluble materials. Solvents were removed by a rotary evaporator to produce a red solid. The red solid was re-dissolved in dry 1,2-dichloroethane under an argon atmosphere and the corresponding diisocyanate (1.1–1.3 eq.) was added. The reaction mixture was heated at 50 °C for overnight and cooled to room temperature. The product mixture was filtered and quickly added trifluoroacetic acid (2 mL), and the mixture was stirred for 6 h at room temperature. Solvents were removed by a rotary evaporator to afford the Cu(I)-catenane, which was re-dissolved in MeOH/CHCl$_3$ (*v/v* 1:1) for recrystallization or direct demetallation.

To obtain the metal-free [2]catenane, a solution of the Cu(I)-catenane in 10 mL MeOH/CHCl$_3$ (*v/v* 1:1) or CH$_2$Cl$_2$ was added 2 mL of neat ethylenediamine (30 mmol), and the mixture was stirred for 5 min, followed by addition of CHCl$_3$ or CH$_2$Cl$_2$ (3 mL) and water (3 mL). The yellow organic phase was separated, and another aliquot of ethylenediamine (1 mL, 15 mmol) was added and stirred for another 5 min to give a colorless solution. Additional CHCl$_3$ or CH$_2$Cl$_2$ (3 mL) and water (3 mL) were added, and the organic phases were combined, washed by water (2 × 5 mL) and dried over Na$_2$SO$_4$. The solvent was removed using a rotary evaporator and the residue was purified by solvent washing or chromatography.

## Computational details

Empirical structures of the 1:1 and 1:2 catenane complexes with anions that showed significant binding (i.e. SO$_4^{2-}$, H$_2$PO$_4^-$, CH$_3$COO$^-$ and Cl$^-$) were generated using the Avogadro software[63], and refined using density functional theory (DFT) calculations performed with Gaussian 16 (revision A.03)[64]. Geometry optimizations and single-point energy calculations were carried out using the spin-restricted hybrid density functional B3LYP[65–67], and the 6–31 G (d,p) basis set was employed for all atoms. The solvation effects of DMSO ($\varepsilon$ = 46.826) were taken into account using the conductor-like polarizable continuum model (CPCM)[68,69]. During the structural optimization step, a convergence criterion of $10^{-8}$ for the density matrix was applied. Vibrational analyses were conducted to validate the optimized structures.

## Data availability

All processed experimental data that support the findings of this study are available within the main text and its Supplementary Information, including synthesis of compounds and characterization, experimental details and data of cation and anion binding studies, computational studies, guest-induced mechanostereochemical switching and X-ray crystallography). X-ray crystallographic data for the structure **C**-TBA$_2$SO$_4$ has been deposited at the Cambridge Crystallographic Data Centre (CCDC), under the deposition number CCDC 2245688. The data can be obtained free of charge via https://www.ccdc.cam.ac.uk/structures/. Coordinates of the optimized structures in this study are provided in the Source Data file. Additional raw data of this study are available from the corresponding author upon request. Source data are provided with this paper.

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

## Acknowledgements

This work is supported by the CAS-Croucher Funding Scheme for Joint Laboratories, the Collaborative Research Fund (C7075-21G to H.Y.A.) and General Research Fund (17300723 to H.Y.A.) from the Research Grants Councils of Hong Kong. Y.Y. is a recipient of the Postgraduate Scholarship from The University of Hong Kong. We acknowledge UGC funding administered by The University of Hong Kong for support of the Electrospray Ionisation Quadrupole Time-of-Flight Mass Spectrometry Facilities under the support for Interdisciplinary Research in Chemical Science.

## Author contributions

Y.Yao: conceptualization, methodology, formal analysis, investigation, writing – original draft, and visualization; Y.C.Tse: methodology, formal analysis, and writing – review & editing; S.K.-M.Lai: investigation; Y.Shi: formal analysis and visualization; K.-H.Low: investigation; H.Y.Au-Yeung: conceptualization, formal analysis, writing – original draft, writing – review & editing, visualization, supervision, project administration, and funding acquisition.

## Competing interests

The authors declare no competing interests.
