## [Peer review file · Nature Communications]

Dynamic mechanostereochemical switching of a conformationally flexible [2]catenane controlled by specific ionic guestsREVIEWER COMMENTS

Reviewer #1 (Remarks to the Author):

In this article the authors report a [2]catenane ion host system, containing bipy and urea functionalities. The synthesis of such a complex host is impressive in its efficiency, although not especially novel as derived from a Sauvage type Cu passive template. The interlocked product suitably characterized, including using 2D/NOESY nmr technique, as well as via Xray crystallography. The authors mention that Zn(II) binding does not occur, which is surprising – a more thorough investigation of the cation binding properties and more general selectivity (other transition metal ions, eg Cu(II)) should be included, perhaps backed up by calculation to rationalize the selectivity.

The anion binding properties were investigated by NMR titrations. The data suggests that sulfate binds within the catenane binding site, as might be expected based on the extensive work in this area of Beer and others. Sulfate selectivity is observed, but not full discussed – for example why does phosphate bind more weakly, despite the higher charge? NMR studies suggest that the rings undergo a rotation with respect to one another, from the initial co-conformation, in order to position the urea motifs ready for convergent binding to the sulfate. Binding of sulfate induced a tilted orientation of the macrocycles, and helical chirality, which is an unusual and notable concept. Successive addition of Cu(I) and sulfate, along with the corresponding competitive ligands, allowed for sequential switching of the catenane conformation.

This is a well conducted study, on a fairly unusual type of catenane receptor. However, given the precedent for mechanical conformation switching with ions (dating back to Sauvage's rotaxane shuttles, ref 58, and for anions in catenanes this has been known for some time e.g. Chem. Commun., 2008, 1281-1283). With the now extensive literature on ion recognition using interlocked molecules, this does not feel like a significant step forwards in the field, albeit a well conducted piece of work bringing together many interesting concepts such as switching, chirality, binding selectivity. Nevertheless, on balance I cannot recommend publication in Nature Communications – the work would be more suitable for a more specialized chemistry journal.

Reviewer #2 (Remarks to the Author):

In this manuscript, Au-Yeung and coworkers demonstrated that, attributed to their large-amplitude co-conformational changes, flexible [2]catenane decorated with different binding sites could serve as receptor for both cation and anion species through dynamic co-conformational switching. Although the selective recognition of either cations and anions using MIMs receptors have been already reported, the study described in this manuscript highlights the great potential of MIMs as adaptive synthetic receptors,

thus publication is strongly recommended. However, some key issues as listed below should be addressed;

1. For the cation binding screening, why didn't the authors try other cations, particularly ones might have strong affinities with the bpy units? In addition to Cu⁺, only another two cations were tested, definitely it is not enough;
2. In the case of the anion binding screening, according to the solution state titration experiments, the binding stoichiometry was determined to be 1:2. However, in the solid-state structure of the complex, the binding stoichiometry is 1:1, how to explain such difference?
3. For ion sensing study, upon the selective bind with cations or anions, there are usually specific signals such as fluorescence changes as the outputs. Thus for the system described in this manuscript, it will be very attractive if different optical outputs would be achieved upon the binding with cation and anion;
4. The mechanically helically chiral feature of the sulfate catenane complex is quite interesting. This reviewer wonders if the TBA cation is replaced by a chiral counter cation, whether will it induce the formation of one selected chiral co-conformer over the other one?
5. Additional computational study are necessary to provide additional supports for the selectivity towards sulfate over other anions;
6. This reviewer is quite curious that what will happen when the [2]catennane is mixed with Cu₂SO₄, whether will it lead to the formation of an interesting framework?
7. According to the molecular packing of the sulfate catenane complex, it will be possible to form linear supramolecular polymers when bis-sulfonates are complexed with the [2]catenane, maybe the authors could use such design strategy for the construction of novel switchable supramolecular polymers.

Reviewer #3 (Remarks to the Author):

The authors report on a nice mechanically interlocked system with an ability to switch between two conformations upon binding to Cu or SO₄²⁻.

The presented work is well done and the analysis of the system and its binding abilities towards various cations and anions are examined based on the NMR and UV-vis absorption titrations experiments. Model catenane C' with only two binding sites for SO_4^{2-} was also prepared. The citation of the contribution of other authors in the field is very appreciated.

Concerning the molecular design and the synthesis of the present system, there is no real challenge as all is based on well-established chemistry and similar catenanes have been already reported. The preparation of the catenanes reported therein is based on the Cu-templated synthesis developed in 1983 by Sauvage and co-workers. The use of urea motif as binding site for anions such as SO_4^{2-} has been also already reported. Additionally, the high affinity of the urea motif for sulfate was already demonstrated (reference 26). In the cited work, $\log K = 4.9$ was found for the 1:1 complex. That's why it is not really surprising that the urea units in the catenane structure is able to bind so strongly sulfate anions when the system undergoes the structural reorganization. This facility of the system to reorganize the cavity upon cation or anion binding is indeed elegant. However, the following statement is inappropriate: "These findings suggest the mode of sulfate binding by C features a cooperative mechanical chelation of a sulfate anion by two urea groups from the two interlocked rings, and more importantly highlights the critical role of the mechanical flexibility for efficient, large- amplitude co-conformational rearrangement of the binding motifs despite of the rigidity of the covalent macrocyclic skeleton." Indeed, considering K_2/K_1 , a value of 0.17 is obtained for sulfate binding in 5% $\text{D}_2\text{O}/\text{DMSO-d}_6$. This value is lower than 0.25 (statistical binding over two equivalent binding sites) and therefore evidencing a negative cooperativity. As discussed by the authors, one has the feeling that they mention a positive cooperativity which is not the case. For AcO^- and H_2PO_4^- , negative cooperative effects are also obvious based on the relative K_A values. A proper discussion on the negative cooperativity should added. Moreover, thermodynamic parameters should be also reported. It is essential to see the relative entropic and enthalpic contributions. This is also needed to understand the negative cooperativity.

The new and interesting element of the presented studies concerns the switching between achiral and chiral co-conformation when sulfate is bind. The 1D and 2D NMR measurements were used to show the relative orientation of two macrocycles in catenane structure. However, the racemization process if any should be investigated by variable temperature studies. The system is rather flexible and racemization within the NMR timescale is expected.

It would be interesting to know if the length of a linker between two urea units influences such conformational switching observed upon cation/anion binding. It is a pity that the authors did not prepare catenanes with a linker of another length to check if the relative orientation of smaller or bigger macrocycles will still induce the chirality of such catenane-based receptor after sulfate anion binding?

During the synthetic procedure, the authors used ethylenediamine as an alternative to toxic cyanide when Cu needs to be extracted. So why during the cycle with sequential addition and removal of sulfate and Cu(I) guests (supplementary Fig. 40) TMACN was used instead of ethylenediamine?

Globally, the manuscript presents very interesting studies on the conformational switching behavior of a catenane-based receptor upon cation and/or anion binding leading to achiral/chiral co-conformations of

the system. The manuscript is well written and the experimental work is of high quality. The presented results could be of interest for the Nature Communications readers and I'm in favor of acceptance. The work should be however completed as indicated in the present report.

Reviewers' comments are laid out below in *italicized font* and our answers are marked in blue.

Reviewer #1

In this article the authors report a [2]catenane ion host system, containing bipy and urea functionalities. The synthesis of such a complex host is impressive in its efficiency, although not especially novel as derived from a Sauvage type Cu passive template. The interlocked product suitably characterized, including using 2D/NOESY nmr technique, as well as via Xray crystallography. The authors mention that Zn(II) binding does not occur, which is surprising – a more through investigation of the cation binding properties and more general selectivity (other transition metal ions, eg Cu(II)) should be included, perhaps backed up by calculation to rationalize the selectivity.

We sincerely thank the reviewer for the careful evaluation of our work. We agree that the apparently weak Zn(II) binding is perhaps surprising and requires further investigation. To begin with, we chose 6,6'-dimethyl-2,2'-bipyridine (**DMbpy**) as a model ligand to investigate the Zn(II) binding by ^1H NMR titration. Our results showed that the Zn(II)-bpy coordination is strongly dependent on the solvent. While no obvious spectral change was observed when the titration was conducted in $\text{DMSO-}d_6$, a new set of bipyridine signals in slow exchange with those of the free ligand was observed when the titration was conducted in CD_3CN , suggesting that the Zn(II)-bpy coordination is likely kinetically controlled and is very slow in $\text{DMSO-}d_6$ (Figure R1). For the limited solubility of catenane **C**, for which NMR studies can only be conducted in DMSO but not acetonitrile, we thus propose that one reason of the lack of significant binding observed for **C** to Zn^{2+} is due to a kinetic effect in the DMSO solvent.

Figure R1. Partial ^1H NMR spectra (500 MHz, 298 K) of **DMbpy** in the presence of 0 eq. (top), 0.2 eq., 0.3 eq., 0.5 eq., 1 eq. and 10 eq. (bottom) of Zn^{2+} in (a) $\text{DMSO-}d_6$ and (b) CD_3CN , respectively. Signals from the Zn^{2+} -complexed **DMbpy** are highlighted in orange.

On the other hand, similar kinetic effects have also been reported for the direct metalation of a bpy-derived molecular knot, in which the identity of the metals and counterions were found to influence the degree of the metalation (*J. Am. Chem. Soc.*, 2019, **141**, 3952). In that reported study, it is proposed that due to the presence of different (co)conformations of the bpy-derived MIM, rearrangement of the ligand strands for binding the metal in the “correct” geometry could result in a kinetic effect. Considering that catenane **C** is also (co)conformationally flexible, a similar rearrangement kinetic may also be part of the reason of why no significant zinc binding was observed in our case.

In addition to Cu^+ , Zn^{2+} and Na^+ described in the original version of the manuscript, binding of catenane **C** towards other main group and transition metals including Li^+ , Fe^{2+} , Co^{2+} , Ni^{2+} , Cu^{2+} , Ag^+ and Cd^{2+} were also studied by ^1H NMR, UV-Vis and HR-MS, and no observable binding was found for Li^+ , Fe^{2+} , Co^{2+} , Ni^{2+} , and Cd^{2+} (Figures R2–R4). Reduction of Cu^{2+} upon adding the divalent metal to a DMSO solution of the catenane was observed, resulting in the formation of the Cu(I)-catenane complex $[\text{CuC}]^+$ as shown by ^1H NMR (Figure R2h) and UV-Vis (Figure R3b). The Cu(II) reduction is consistent that observed in a related catenane system (*Inorg. Chem. Front.*, 2016, **3**, 348) and also the catenand effect that stabilizes the metal in a low oxidation state (*J. Am. Chem. Soc.*, 1989, **111**, 7791). For ^1H NMR titration of Ag^+ to the catenane, a gradual shift of the proton resonances of the catenane was observed, suggesting the formation of a new species that is fast exchanging with **C** (Figure R4). Considering the characteristic slow ligand exchange due to catenand effect, this fast exchange would be quite unusual if the Ag^+ is fully encapsulated in the catenane core in a 4-coordinate fashion. Nevertheless, subsequent addition of stoichiometric amount of Cu^+ to the Ag^+ -containing titration mixture resulted in complete conversion to $[\text{CuC}]^+$, showing that the Ag^+ binding to **C** is much weaker, and the binding is selective to Cu^+ over Ag^+ . Nevertheless, a more thorough investigation on the coordination chemistry of bpy-derived catenanes towards Ag^+ will be necessary to fully elucidate these observations, and these studies may require bpy-based catenanes that are more soluble in other solvents due to the possible solvent effects and other complications as encountered in the case of Zn^{2+} .

The manuscript has been revised with the addition of these discussions and relevant citations. The additional data and experimental details are also provided in the updated Supplementary Information.

Figure R2. Partial ^1H NMR spectra (500 MHz, 298 K, $\text{DMSO-}d_6$) of a 1 mM solution of **C** in the presence of 1 eq. of different cations. Signals from H_{bpy} and H_{dpm} are highlighted in blue and pink respectively.

Figure R3. UV-Vis spectra of 125 μM solutions of **C** in DMSO at 298 K in the presence of 0 to 10 eq. of (a) Cu^+ , (b) Cu^{2+} (compared with that in the presence of 1 eq. of Cu^+), (c) Ag^+ , (d) Li^+ , (e) Na^+ , (f) Fe^{2+} , (g) Co^{2+} , (h) Ni^{2+} , (i) Zn^{2+} and (j) Cd^{2+} .

Figure R4. Partial ^1H NMR spectra (500 MHz, 298 K, $\text{DMSO-}d_6$) of a 1 mM solution of **C** in the presence of 0 to 2 eq. of Ag^+ , followed by sequential addition of 0.3 eq. to 1 eq. Cu^+ , compared with that of a 1 mM solution of $[\text{CuC}]^+$ (bottom). Signals from H_{bpy} and H_{tpm} are highlighted in blue and pink respectively.

The anion binding properties were investigated by NMR titrations. The data suggests that sulfate binds within the catenane binding site, as might be expected based on the extensive work in this area of Beer and others. Sulfate selectivity is observed, but not full discussed – for example why does phosphate bind more weakly, despite the higher charge? NMR studies suggest that the rings undergo a rotation with respect to one another, from the initial co-conformation, in order to position the urea motifs ready for convergent binding to the sulfate. Binding of sulfate induced a tilted orientation of the macrocycles, and helical chirality, which is an unusual and notable concept. Successive addition of Cu(I) and sulfate, along with the corresponding competitive ligands, allowed for sequential switching of the catenane conformation.

We thank for the comment and agree that the phosphate binding needs more careful investigation. Because of the high basicity and hygroscopic nature of phosphate, ^1H NMR titration experiment for phosphate binding to the catenane was repeated using a new batch of thoroughly dried $\text{DMSO-}d_6$ and freshly prepared phosphate (from phosphoric acid and tetrabutylammonium hydroxide) (Figure R5). Water content of the $\text{DMSO-}d_6$ for all previous titration is $\sim 0.2\%$ as estimated by ^1H NMR. Upon addition of PO_4^{3-} , the urea NH protons were found disappeared, suggesting that the urea NH may have been deprotonated by the strongly basic phosphate. Further addition of up to 5 eq. of phosphate resulted in a new set of upfield-shifted, broad and overlapped signals, suggesting the presence of multiple exchanging species, likely with the urea NH deprotonated, that may be associated with the phosphate/hydrogen phosphate in the solution. As a result, due to the possible presence of a (de)protonation equilibrium, the complex binding equilibrium involving multiple species derived from the catenanes and phosphate, as well as the difficulty in further analyzing the broad, overlapped and exchanging signals, we believe that interactions of the catenane host with phosphate is likely more complicated than a simple host-guest binding, and therefore decide not to report the phosphate

binding at the present time. The main text has been modified accordingly and the corresponding data and experimental details are provided in the revised Supplementary Information.

Figure R5. Partial ¹H NMR spectra (500 MHz, 298 K, molecular sieve-dried DMSO-*d*₆) of **C** (1 mM) in the presence of 0 eq. (top), 0.25 eq., 0.5 eq., 0.75 eq., 1 eq., 1.5 eq., 2 eq., 2.5 eq., 3 eq., 5 eq., 8 eq. and 12 eq. (bottom) of TBA phosphate. Signals from H_{bpy} and H_{dpm} are highlighted in blue and pink respectively.

Reviewer #2

In this manuscript, Au-Yeung and coworkers demonstrated that, attributed to their large-amplitude co-conformational changes, flexible [2]catenane decorated with different binding sites could serve as receptor for both cation and anion species through dynamic co-conformational switching. Although the selective recognition of either cations and anions using MIMs receptors have been already reported, the study described in this manuscript highlights the great potential of MIMs as adaptive synthetic receptors, thus publication is strongly recommended. However, some key issues as listed below should be addressed;

1. For the cation binding screening, why didn't the authors try other cations, particularly ones might have strong affinities with the bpy units? In addition to Cu⁺, only another two cations were tested, definitely it is not enough;

Thank you for the comment and we agree that more metal ions should be tested to establish the cation selectivity. As mentioned above in a reply to Reviewer 1, we have additionally studied the binding of the catenane towards Li⁺, Fe²⁺, Co²⁺, Ni²⁺, Cu²⁺, Ag⁺ and Cd²⁺ by ¹H NMR, UV-Vis and HR-MS. Details of the results can be found in the reply above.

2. In the case of the anion binding screening, according to the solution state titration experiments, the binding stoichiometry was determined to be 1:2. However, in the solid-state structure of the complex, the binding stoichiometry is 1:1, how to explain such difference?

Thank you for the comment and we agree that the stoichiometry of the sulfate binding may seem to be inconsistent at a first sight. In the crystal structure, while the overall catenane-to-sulfate stoichiometry is 1:1, it should be noted that each catenane is bound to two sulfate ions (through hydrogen bonds), and that each sulfate ion is also bridging two catenane molecules to form an infinite 1-dimensional chain. In solution, the catenane complex is fully solvated under the experimental conditions, and hence formation of such infinite 1-dimensional chain would be unlikely, and the solvated catenane complex can indeed be stabilized by interacting with solvent molecules (i.e. DMSO) as well. Moreover, crystal packing and steric effects of the bulky tetrabutylammonium ions are also important to the crystallization process, and hence will be influencing the specific arrangement of the catenane, sulfate and counterion, as well as their stoichiometry, observed in the specific crystal structure. Nevertheless, in both crystalline and solution states, each catenane molecule can bind to two sulfate ions using the urea groups at the periphery, and hence resulting the observed binding stoichiometry.

3. For ion sensing study, upon the selective bind with cations or anions, there are usually specific signals such as fluorescence changes as the outputs. Thus for the system described in this manuscript, it will be very attractive if different optical outputs would be achieved upon the binding with cation and anion;

We thank the reviewer for the interesting insight. In addition to the UV-Vis data provided in the previous version of the manuscript, we have additionally studied the fluorescence emission of the catenane and its Cu⁺ and sulfate complexes (Figure R6). For a DMSO solution of C, excitation at 270 nm was found to result in two emission peaks at 350 nm and 460 nm. A decrease in the normalized emission intensity at 460 nm was observed upon addition of Cu⁺ or SO₄²⁻ ions. Yet, such non-specific changes in the emission may not be useful to distinguish the binding of the different ions. A possible direction to engineer specific optical changes would be to incorporate strongly absorbing chromophores and/or emitting fluorophores at specific positions of the catenane, such that the

different co-conformations resulted from the binding of different ions may induce a different arrangement or interactions of the optically active units to result in different optical outputs, and it will be our next objective to investigate additional synthetic strategies, as well as detailed co-conformational and structural studies for achieving these goals in future studies. The new fluorescence data and the experimental details are included in the updated Supplementary Information.

Figure R6. Normalized fluorescence spectra of the 125 μM solution of C in DMSO at 298 K in the presence of (a) 0 to 2.0 eq. of Cu⁺ and (b) 0 to 20 eq. of SO₄²⁻.

4. *The mechanically helically chiral feature of the sulfate catenane complex is quite interesting. This reviewer wonders if the TBA cation is replaced by a chiral counter cation, whether will it induce the formation of one selected chiral co-conformer over the other one?*

We thank the reviewer for the suggestion. To study the effect of a chiral counter cation in the sulfate, it would be ideal to directly use a sulfate salt with a chiral cation in the binding study. The commercially available quinine sulfate was thus first chosen to replace TBA₂SO₄ in the binding study, but quinine sulfate salt was found to display a very low solubility in DMSO-*d*₆ for the titration. Since the catenane also displays limited solubility except in DMSO, we turned our attention to the addition of a large excess of a chiral cation to a solution of the catenane-sulfate complex to see if the chiral cation would have any influence, from possible ion-pairing, on the chirality of the complex. By CD and ¹H NMR spectroscopies (Figure R7), no significant spectral changes were observed upon addition of an excess of *N,N,N*-trimethyl-L-valine methyl ester ammonium hexafluorophosphate to a DMSO solution of C in the presence of 3 eq. of TBA₂SO₄. While a specific stereoisomer of the catenane-sulfate complex could in principle be stabilized by a chiral cation, synthesis of other chiral sulfate salts with an appropriate solubility will be necessary for further investigation. Nevertheless, these preliminary data and the experimental details are provided in the updated Supplementary Information.

Figure R7. (a) CD spectra of **C** (125 μM in DMSO, 298 K) with 10 eq. of TBA sulfate in the presence of 0 to 50 eq. of chiral cation L^+ , and partial ^1H NMR spectra (500 MHz, 298 K, $\text{DMSO-}d_6$) of **C** (1 mM) with 3 eq. of TBA sulfate (b) before and (c) after the addition of 6 eq. of L^+ .

5. Additional computational study are necessary to provide additional supports for the selectivity towards sulfate over other anions;

We appreciate the reviewer's suggestion and have included additional computational simulations. To simulate the anion complexes of the catenane, empirical structures of the 1:1 and 1:2 catenane complexes with anions that showed significant binding (i.e. SO_4^{2-} , H_2PO_4^- , CH_3COO^- and Cl^-) were first optimized and refined using density functional theory (DFT) calculations performed with Gaussian 16 (revision A.03), followed by the geometry optimizations and single-point energy calculations using the spin-restricted hybrid density functional B3LYP with a 6-31G (d,p) basis set for all the atoms. The solvation effects of DMSO ($\epsilon = 46.826$) were taken into account using the conductor-like polarizable continuum model. During the structural optimization step, a convergence criterion of 10^{-8} for the density matrix was applied and vibrational analyses were conducted to validate the optimized structures. The free energy of binding was obtained by the difference of the free energies of the unbound and complexed species. The computed association constants were found comparable to our experimental data, and all the binding were found to be enthalpy driven with a negative entropy (SO_4^{2-} : $\Delta H = -74.06 \text{ kcal}\cdot\text{mol}^{-1}$, $T\Delta S = -30.73 \text{ kcal}\cdot\text{mol}^{-1}$; H_2PO_4^- : $\Delta H = -24.46 \text{ kcal}\cdot\text{mol}^{-1}$, $T\Delta S = -16.11 \text{ kcal}\cdot\text{mol}^{-1}$; CH_3COO^- : $\Delta H = -26.29 \text{ kcal}\cdot\text{mol}^{-1}$, $T\Delta S = -17.51 \text{ kcal}\cdot\text{mol}^{-1}$; Cl^- : $\Delta H = -12.02 \text{ kcal}\cdot\text{mol}^{-1}$, $-8.33 \text{ kcal}\cdot\text{mol}^{-1}$).

The energy minimized structures showed that the different anions could interact with different parts of the catenane apart from the urea (Figure R8). For example, in the catenane complex with H_2PO_4^- , hydrogen bonds between the OH of the anion and the ureido carbonyl O, and C-H \cdots O hydrogen bonds between the P=O of the anion and the picolylic CH are observed (Figure R8b). In addition, inter-ring interactions such as N-H \cdots O hydrogen bonds between the different urea groups from the two interlocked rings are also found in the catenane complexes of H_2PO_4^- (Figure R8b), Cl^- (Figure

R8d), as well as the 1:1 sulfate complex (Figure R8a, left). These diverse structures obtained from the computational studies hence suggest that these other interactions involving other parts of the catenane, which may be less obvious than the hydrogen bonds between the anions and the urea groups, are also important in the overall stability of the anion complex. In other words, the observed anion selectivity is likely a result of the overall balance of the structural compatibility between the anions and the catenane, as well as the resulting local interactions involving the different catenane co-conformations. The related discussion is added in the revised manuscript, and the computation data and methods of the simulation are provided in the updated Supplementary Information.

Figure R8. Optimized structures of 1:1 (left) and 1:2 (right) complexes of C with (a) SO_4^{2-} , (b) H_2PO_4^- , (c) CH_3COO^- and (d) Cl^- . Hydrogen bonds are highlighted in green. Hydrogen atoms of C-H not involved in hydrogen bonding are omitted.

6. This reviewer is quite curious that what will happen when the [2]catennane is mixed with Cu_2SO_4 , whether will it lead to the formation of an interesting framework?

We appreciate the reviewer's suggestion. As Cu_2SO_4 is unstable and not readily available, we instead studied the co-presence of the two ions by ^1H NMR titration of Cu^+ ion to a 1 mM solution of the 1:2 sulfate complex of **C** in $\text{DMSO-}d_6$ (i.e. 3 mM of TBA_2SO_4 , Figure R9). Upon addition of up to 1 mole eq. of Cu^+ (relative to **C**), no significant spectral changes were observed. Broad signals were observed when a further 0.5 eq to 2 eq. of Cu^+ (1.5 eq. to 3 eq. in total, relative to **C**) was added, suggesting the presence of several exchanging species with an exchange rate close to that of the NMR. The co-presence of Cu^+ and SO_4^{2-} ions hence likely resulted in the formation of both complexes, as well as related ion pairs, that are in exchange in solution. Further addition of Cu^+ of up to 6 eq. (relative to **C**) resulted in the emergence of a new set of signals with chemical shifts very close to that of $[\text{CuC}]^+$, consistent with a shifting of the equilibrium towards the Cu(I)-catenane complexes as predicted by Le Chatelier's principle. The experimental data are provided in the updated Supplementary Information.

Figure R9. Partial ^1H NMR spectra (500 MHz, 298 K, $\text{DMSO-}d_6$) of **C** (1 mM) with 3 eq. of TBA sulfate in the presence of 0 eq. to 6 eq. of Cu^+ , compared with that of pure $[\text{CuC}]^+$ (bottom).

7. According to the molecular packing of the sulfate catenane complex, it will be possible to form linear supramolecular polymers when bis-sulfonates are complexed with the [2]catenane, maybe the authors could use such design strategy for the construction of novel switchable supramolecular polymers.

We appreciate the reviewer's suggestion and will study the possibility of catenane binding to sulfonate, bis-sulfonate and related compounds in our future studies. As described in the original version of the manuscript, hydrogen sulfate was found to bind only weakly and showed only slight chemical shift changes upon addition of 10 eq. of hydrogen sulfate, suggesting that the anion binding of the catenane may be very sensitive to even a slight change in the structure and charge of the anionic guest.

Reviewer #3

*The authors report on a nice mechanically interlocked system with an ability to switch between two conformations upon binding to Cu or SO_4^{2-} . The presented work is well done and the analysis of the system and its binding abilities towards various cations and anions are examined based on the NMR and UV-vis absorption titrations experiments. Model catenane **C'** with only two binding sites for SO_4^{2-} was also prepared. The citation of the contribution of other authors in the field is very appreciated.*

We thank the reviewer for the detailed evaluation and supportive comments on our work.

Concerning the molecular design and the synthesis of the present system, there is no real challenge as all is based on well-established chemistry and similar catenanes have been already reported. The preparation of the catenanes reported therein is based on the Cu-templated synthesis developed in 1983 by Sauvage and co-workers. The use of urea motif as binding site for anions such as SO_4^{2-} has been also already reported. Additionally, the high affinity of the urea motif for sulfate was already demonstrated (reference 26). In the cited work, $\log K = 4.9$ was found for the 1:1 complex. That's why it is not really surprising that the urea units in the catenane structure is able to bind so strongly sulfate anions when the system undergoes the structural reorganization. This facility of the system to reorganize the cavity upon cation or anion binding is indeed elegant.

We thank the reviewer for the comments and support on the work. We also agree that the Cu(I) template and dynamic urea formation, as well as the use of urea for anion binding are all well-established.

Yet, we would also like to highlight that a successful catenane synthesis may involve considerations more than just the template and ring-closing reaction. Related to another question concerning the possible effects of the linker of the catenane (see below), the dimension, flexibility and likely other structural factors of the precursors, which control how the reactive ends are connected during the macrocycle formation, are actually another important factors that need to be taken into account for a successful and high-yielding catenane synthesis. In fact, we have also tried the synthesis of other similar urea-containing catenanes from the Cu(I)-bis(bpy) complex $[Cu(1)_2]^+$ using other diisocyanate precursors, and found that flexible precursors (e.g. hexamethylene diisocyanate and octamethylene diisocyanate) would result in the formation of a single macrocycle from a [2+2] macrocyclization instead of a [2]catenane (Scheme R1). While some other more rigid precursors (e.g. 1,3-xylylene diisocyanate) could result in [2]catenane formation from a [1+1] macrocyclization, but the [2]catenane yields are rather low (ca. 8%) with the concomitant formation of other non-interlocked

side-products. Not only the purification and isolation of pure [2]catenane sample is more complicated, but also the ion binding studies are more challenging with limited amount of materials. Similar observations on the complication related to precursor connectivity in catenane synthesis have also been reported in a few other literatures (e.g. *J. Am. Chem. Soc.*, 2005, **127**, 12612; *Chem. Commun.*, 2014, **50**, 2857; *J. Am. Chem. Soc.*, 2014, **136**, 13142; *Chem. Commun.*, 2019, **55**, 6169).

To avoid the connectivity problem, we have also tried to first synthesize a single bpy macrocycle from precursor **1** and the diisocyanate, from which a pseudorotaxane complex can be formed by reacting with another molecule of precursor **1** using the Cu^+ template, and subsequent ring-closing with another diisocyanate to form the second interlocked ring and give the [2]catenane. However, the obtained macrocycle displayed only limited solubility and formation of the Cu^+ -templated pseudorotaxane complex was unsuccessful. Although current studies on the impact of the structural properties of the precursors on catenane synthesis are still scarce and scattered, catenane synthesis may be perhaps not as straightforward as one would have expected, even when well-established templates and ring-closing reactions are used. More comprehensive and systematic investigations on the synthesis of these related catenanes are underway in our laboratory, and the results will be published in the future.

Scheme R1. Synthesis of [2]catenane **C''** and figure-of-eight [Cu(**F8**)]PF₆.

However, the following statement is inappropriate: “These findings suggest the mode of sulfate binding by **C** features a cooperative mechanical chelation of a sulfate anion by two urea groups from the two interlocked rings, and more importantly highlights the critical role of the mechanical flexibility for efficient, large- amplitude co-conformational rearrangement of the binding motifs despite of the rigidity of the covalent macrocyclic skeleton.” Indeed, considering K_2/K_1 , a value of 0.17 is obtained for sulfate binding in 5% D₂O/DMSO-*d*₆. This value is lower than 0.25 (statistical binding over two equivalent binding sites) and therefore evidencing a negative cooperativity. As discussed by the authors, one has the feeling that they mention a positive cooperativity which is not the case. For AcO⁻ and H₂PO₄⁻, negative cooperative effects are also obvious based on the relative K_A values. A proper discussion on the negative cooperativity should be added. Moreover, thermodynamic parameters should be also reported. It is essential to see the relative entropic and enthalpic contributions. This is also needed to understand the negative cooperativity.

We sincerely thank the reviewer for the comment and agree that the wordings in the original version of the manuscript could be misleading. We previously chose the word “cooperative” to describe the chelation of one sulfate anion by the catenane, because the binding of the two urea groups from the two different rings is expected to be cooperative due to their proximity as a result of the interlocking. We did not intent to indicate the binding of the two sulfate ions are positively cooperative, and agree that the previous version could lead to confusion. The text has been revised to avoid the confusion.

For the negative cooperativity of the sulfate binding, since **C** is an electrically neutral [2]catenane, binding of the second anionic guest will result in an unfavourable electrostatic repulsion (especially for the dianionic sulfate), and hence can explain the observed negative cooperativity (*J. Am. Chem. Soc.*, 2014, **136**, 7505). We have also conducted additional binding experiments to obtain the enthalpy and entropy change of the sulfate binding by Van’t Hoff analysis by performing a series of titration experiments to obtain binding constants at different temperatures (we were not able to conduct ITC due to an incompatibility issue with the DMSO solvent). The results showed that the overall sulfate binding in 5% D₂O/DMSO-*d*₆ is characterized by a ΔH of $-23(\pm 3)$ kcal·mol⁻¹ and a ΔS of $-41(\pm 9)$ cal·mol⁻¹·K⁻¹, showing that the sulfate binding is enthalpy driven due to the favourable hydrogen bonds (and other interactions) between the sulfate and the urea groups. The negative entropy, on the other hand, may indicate the decrease in the co-conformational flexibility of the catenane upon binding to the sulfate ions.

The stepwise enthalpy and entropy changes obtained by linear fitting also suggested that both the first and second sulfate binding in 5% D₂O/DMSO-*d*₆ are enthalpy driven ($\Delta H_1 = -12(\pm 3)$ kcal·mol⁻¹, $\Delta S_1 = -21(\pm 8)$ cal·mol⁻¹·K⁻¹; $\Delta H_2 = -11(\pm 2)$ kcal·mol⁻¹, $\Delta S_2 = -21(\pm 4)$ cal·mol⁻¹·K⁻¹). While the more favourable first sulfate binding and negative cooperativity can be ascribed to the electrostatic repulsion as mentioned above, the similar entropy changes for the first and second sulfate binding suggest that the monosulfate complex of the catenane are probably more flexible, and as such the second sulfate binding is also accompanied by a negative entropy change. In fact, since sulfate binds at the peripheral positions of the catenane, and that the anion may also interact favourably with other parts of the catenane in addition to the urea (as shown by DFT simulation, see above for details), such a flexible monosulfate complex may not be that surprising. The decrease in the entropy in the second sulfate binding, as opposed to those other well-preorganized ditopic hosts, may hence also contribute to the overall negative cooperativity observed in our system. On the other hand, we have also performed DFT calculations on both the 1:1 and 1:2 catenane-sulfate complexes, as well as those of H₂PO₄⁻, CH₃COO⁻ and Cl⁻ to further understand the binding. The stepwise association constants obtained from the simulation agree well with the experimentally measured association constants. For the energy-minimized structure of the monosulfate complex, apart from the hydrogen bonds between the sulfate and the urea, other interactions involving other parts of the catenane can also be observed

(e.g. hydrogen bonds between two urea groups from the two interlocked rings). The overall stability of the monosulfate complex is hence likely an overall contribution of all these non-covalent interactions in addition to that of the sulfate-urea hydrogen bonds. The related discussion is added in the revised manuscript, and the data and methods of the experiments and simulation are provided in the updated Supplementary Information.

The new and interesting element of the presented studies concerns the switching between achiral and chiral co-conformation when sulfate is bind. The 1D and 2D NMR measurements were used to show the relative orientation of two macrocycles in catenane structure. However, the racemization process if any should be investigated by variable temperature studies. The system is rather flexible and racemization within the NMR timescale is expected.

We thank the reviewer for the suggestion, and variable temperature ^1H NMR analysis was performed to study the racemization of the 1:2 sulfate complex of the catenane in $\text{DMSO-}d_6$ (Figure R10). By increasing the temperature from 298 K to 358 K, the picolyl methylene protons were found to coalesce at about 318 K, corresponding to an energy barrier ΔG^\ddagger of $15.7 \text{ kcal}\cdot\text{mol}^{-1}$.

Figure R10. Partial ^1H NMR spectra (500 MHz, $\text{DMSO-}d_6$) of **C** (1 mM) in the presence of 3 eq. of TBA sulfate obtained at temperature from 298 K to 358 K. Diastereotopic splitting of H_1 is highlighted in orange. Coalescence temperature of signal H_1 $T_c = 318 \text{ K}$, $\Delta G^\ddagger = 15.7 \text{ kcal}\cdot\text{mol}^{-1}$.

It would be interesting to know if the length of a linker between two urea units influences such conformational switching observed upon cation/anion binding. It is a pity that the authors did not prepare catenanes with a linker of another length to check if the relative orientation of smaller or bigger macrocycles will still induce the chirality of such catenane-based receptor after sulfate anion binding?

We appreciate the reviewer's comment and suggestion. As mentioned above, synthesis of related urea-derived catenanes may not be as straightforward as we would have expected due to various reasons including solubility and the different possible connectivity of the reactive ends during macrocycle formation. We are currently exploring other synthetic strategies and will report their anion binding and possible chirality induction in the future.

During the synthetic procedure, the authors used ethylenediamine as an alternative to toxic cyanide when Cu needs to be extracted. So why during the cycle with sequential addition and removal of sulfate and Cu(I) guests (supplementary Fig. 40) TMACN was used instead of ethylenediamine?

The reason of using cyanide in the NMR experiment is due to its stronger Cu^+ -coordinating ability, and hence higher efficiency in the Cu^+ removal from the catenane. The resulting Cu(I) cyanide complexes are also diamagnetic that would not interfere with the subsequent NMR studies. If ethylenediamine is used, not only a larger amount of ethylenediamine will be required which could interfere with the later Ba^{2+} ions and sulfate switching, but also the paramagnetic Cu(II) ethylenediamine complexes produced will result in paramagnetic broadening of the resulting ^1H NMR spectra (*Eur. J. Inorg. Chem.*, 2022, e202200271). In the synthesis, on the other hand, as the amount of the catenane involved is relatively large, it is more advantageous to use the non-toxic ethylenediamine as the demetallating agent due to safety concerns.

Globally, the manuscript presents very interesting studies on the conformational switching behavior of a catenane-based receptor upon cation and/or anion binding leading to achiral/chiral co-conformations of the system. The manuscript is well written and the experimental work is of high quality. The presented results could be of interest for the Nature Communications readers and I'm in favor of acceptance. The work should be however completed as indicated in the present report.

We would like to again appreciate the reviewer's supportive comments and suggestions on the revision of our manuscript.

REVIEWERS' COMMENTS

Reviewer #1 (Remarks to the Author):

The authors have presented a detailed revision, that in my view addresses my technical queries and that of the other reviewers. Whilst I originally had reservations about novelty, following this round of revision which has significantly improved the manuscript I am willing to revise my original recommendation and now recommend publication.

Reviewer #2 (Remarks to the Author):

This is a revised manuscript resubmitted by Ho Yu Au-Yeung and coworkers. In this manuscript, the authors realized the dynamic mechanostereochemical switching of a co-conformationally flexible [2]catenane controlled by specific ionic guests. This is a very important contribution to the field of mechanically interlocked systems. I have checked the revised draft carefully. I can tell that the authors have revised the manuscript very well according to the comments of reviewers. Definitely, the quality of the current version has been dramatically improved. So I strongly recommend the current version to be accepted and published in Nature Communications as it.

Reviewer #3 (Remarks to the Author):

The authors have prepared the revised version of the manuscript. It is very appreciated that many additional experiments have been performed to confirm and/or explain different behavior of the catenane receptor towards various cations and anions. The thermodynamic parameters supported by DFT calculations are helpful to better understand the observed selectivity in anion binding.

The additional explanations and comments provided by the authors in answer to the referee's questions or added to the main text of the manuscript, allow for a better global view of the described work.

To summarize, the manuscript presents very nice studies on the conformational switching behavior of an adaptive catenane-based receptor upon cation and/or anion binding leading to achiral/chiral co-conformations of the system. The work is supported by a full set of experiments, DFT calculations and the appropriate discussions. The authors took time and care to provide the necessary explanations to

the referees. In my opinion, the revised version of a manuscript is now suitable for publication and the presented results will be of interest for the Nature Communications readers.